

# A numerical process study on the rapid transport of stratospheric air down to the surface over western North America and the Tibetan Plateau

Bojan Škerlak[1], Stephan Pfahl[1,2], Michael Sprenger[1], and Heini Wernli[1]

[1]Institute for Atmospheric and Climate Science, ETH Zurich, Switzerland
[2]Institute of Meteorology, Freie Universität Berlin, Germany

**Correspondence:** Michael Sprenger (michael.sprenger@env.ethz.ch)

**Abstract.** Upper-level fronts are often associated with the rapid transport of stratospheric air along tilted isentropes to the middle or lower troposphere, where this air leads to significantly enhanced ozone concentrations. Only occasionally these plumes of originally stratospheric air can be observed at the surface, because (i) stable boundary layers prevent an efficient vertical transport down to the surface, and (ii) if boundary layer turbulence is strong enough to enable this transport, the

originally stratospheric air mass is strongly diluted by mixing such that only a weak stratospheric signal can be recorded at the surface. Most documented examples of stratospheric air reaching the surface are from mountainous regions. This study investigates two such events, using a passive stratospheric air mass tracer in a mesoscale model to explore the processes that enable the transport down to the surface. The events occurred in early May 2006 in the Rocky Mountains and in mid June 2006 on the Tibetan Plateau. In both cases, a tropopause fold associated with an upper-level front enabled stratospheric air to

enter the troposphere. In our model simulation of the North American case, the strong frontal zone reaches down to 700 hPa and leads to a fairly direct vertical transport of the stratospheric tracer along the tilted isentropes to the surface. In the Tibetan Plateau case, however, no near-surface front exists and a reservoir of high stratospheric tracer concentrations forms at 300-400 hPa, without further isentropic descent. Entrainment at the top of the very deep boundary layer (reaching to 300 hPa over the Tibetan Plateau) and turbulence within the boundary layer fosters downward transport of stratospheric air to the surface.

Interestingly, despite the strongly differing dynamical processes, stratospheric tracer concentrations at the surface reach peak values of 10-20% in both cases, corroborating the potential of deep stratosphere-to-troposphere transport events to significantly influence surface ozone concentrations in these regions.

## 1 Introduction

It is well known that the transport of air masses from the lower stratosphere to the extratropical troposphere, so-called

stratosphere-to-troposphere transport (STT) significantly contributes to the tropospheric ozone budget (e.g., Roelofs and Lelieveld, 1997). Numerous observational studies revealed that enhanced near-surface ozone concentrations episodically occur due to dry filaments of originally stratospheric air descending down to the lower troposphere. This observational evidence is mainly based on in situ aircraft measurements (e.g., Esler et al., 2003; Homeyer et al., 2011), on airborne and ground-based lidar observations



(e.g., Browell et al., 1987; Trickl et al., 2010), and on ozone and water vapor sondes (e.g., Beekmann et al., 1997; Akritidis et al., 2018). Often, such events are dynamically related to tropopause folds that form in intense upper-level fronts (Keyser and Shapiro, 1986; Škerlak et al., 2015). In these folds, clear-air turbulence and diabatic processes related to deep convection and high-level clouds can reduce potential vorticity, which allows the involved air parcels to cross the tropopause and then to

descend down to the lower troposphere along the tilted isentropes equatorward of the upper-level front. This isentropic descent typically goes along with rapid long-range equatorward transport from the location of the fold to the "arrival" in the lower troposphere [see, e.g., Fig. 13 in Wernli and Davies (1997), showing a deep STT event descending from 60°N near Scotland to < 20°N over the Arabian Sea in less than a week].

Whereas the main mechanisms leading to STT events into the middle and lower troposphere are fairly well understood, the

ensuing question whether the descending air can enter the planetary boundary layer (PBL) and reach the surface is considerably more complex. Škerlak et al. (2014) referred to STT reaching into the PBL as "deep STT" and showed, based upon a climatological Lagrangian analysis of STT with ERA-Interim reanalysis data, that deep STT events are comparatively rare and occur preferentially in high-mountain areas in the subtropics (see their Fig. 5). This is in line with in situ measured signatures of deep STT air masses at high-altitude stations (e.g., Davies and Schuepbach, 1994; Stohl et al., 2000; Cristofanelli et al.,

2010; Lefohn et al., 2012). However, a few examples of STT signals at low altitude stations have also been reported (e.g., Akritidis et al., 2010). Note that surface signals of deep STT are usually rather weak with typical ozone enhancements of less than 20 ppbv (e.g., Gerasopoulos et al., 2006). Exceptions are a few documented events at mountain peaks with spectacular increases of surface ozone by up to 100 ppbv [see discussion in Davies and Schuepbach (1994)]. The reasons why deep STT occurs rarely, typically leads to weak ozone signals, and is most frequent in mountainous regions are manifold:

– for most upper-level fronts, the isentropes become flatter in the lower troposphere (see, e.g., Figs. 15 and 16 in Trickl et al., 2016), which impedes a further adiabatic descent;

– the layer beneath this "terminal altitude of isentropic descent" is often characterized by a (very) stable PBL (see same figures in Trickl et al., 2016), sometimes with a capping inversion that effectively protects the surface from any deep STT influence;

– active PBL dynamics (e.g., the formation of a deep convective boundary layer) is required to overcome the stable stratification barrier and to entrain deep STT air masses into the PBL and transport them down to surface by turbulent mixing;

– this turbulent mixing dilutes the stratospheric characteristics of the deep STT air;

– high mountains can intersect stable PBLs, which makes them more easily accessible for deep STT;

– over high-altitude plateaus (e.g., Tibet) extremely high-ranging PBLs can develop due to the strong influence of lower-
stratospheric PV anomalies (Chen et al., 2016).

The pioneering observational study of Jonson and Viezee (1967) provided an early discussion of these aspects and reviewed most of the relevant literature available at that time. These authors suggested four possible transport pathways or mechanisms



for stratospheric air to reach the lower troposphere and (possibly) the surface, as schematically summarised in Fig. 1. Type 1 describes the dissipation of a stratospheric intrusion (here displayed as a tropopause fold) by "general mixing and diffusion into the free troposphere". In a type 2 case, the stratospheric intrusion reaches the top of the PBL and stratospheric air is subsequently mixed down to the ground by "turbulent eddies and convection at the top of the boundary layer". Type 3 refers to a situation where the intrusion couples to a frontal zone near the surface such that the frontal circulation facilitates the transport to the ground [see also Fig. 7 in Bourqui and Trépanier (2010)]. Finally, type 4 is an extension of type 3 in which downdrafts associated with rainshowers or thunderstorms also play an important role. While these four mechanisms are most likely not to be considered mutually exclusive, the schematic nicely depicts the different processes involved and serves as a framework to discuss the deep STT events in this study.

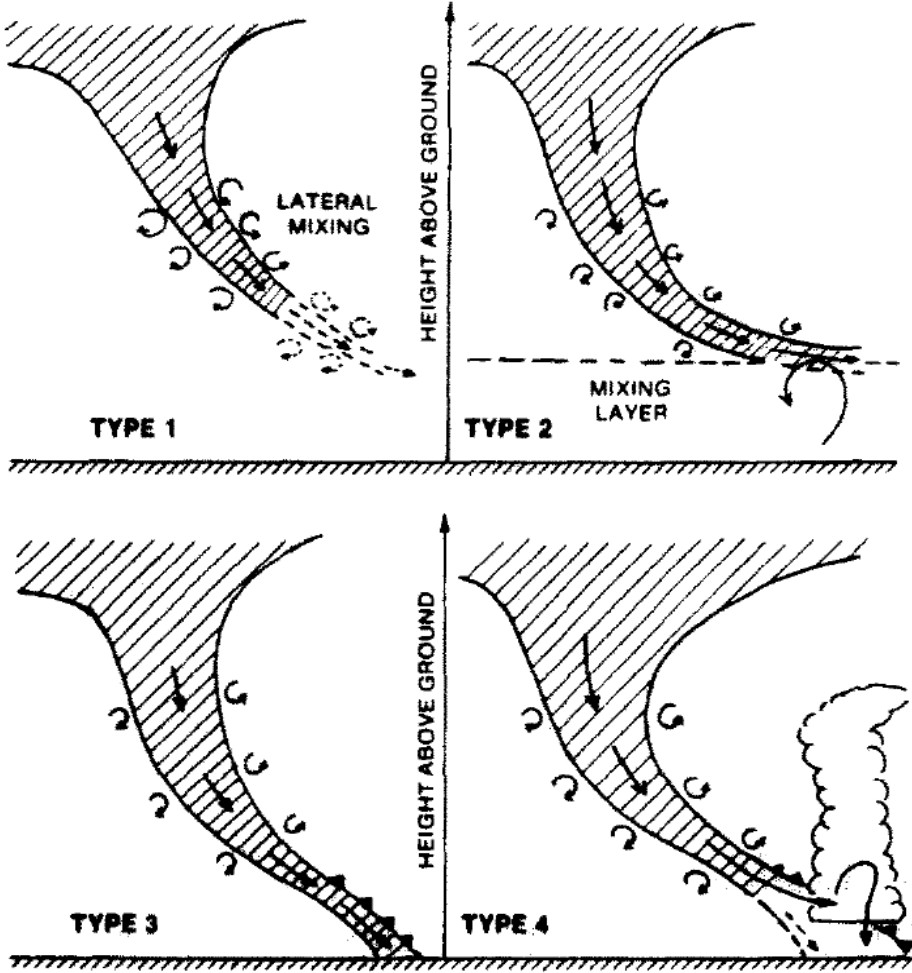

**Figure 1.** Schematic representation of four mechanisms bringing stratospheric air into the lower troposphere and potentially down to the surface [from Johnson and Viezee (1967); their Fig. 10; reproduced with permission from Elsevier].





The discussion so far reveals that deep STT down to the surface involves, in many cases, interactions of tropopause folds, long-range transport, and turbulent PBL dynamics (often in regions with steep topography). These interactions are complex and encompass processes on scales of $> 1000\,\mathrm{km}$ (transport), $\sim 100\,\mathrm{km}$ (tropopause fold), and $< 1\,\mathrm{km}$ (mixing in PBL). In their comprehensive review of stratosphere-troposphere exchange, Stohl et al. (2003) referred to the correct quantification

of mixing of stratospheric with tropospheric air as an unsolved problem. Obviously, the temporal and spatial resolution of global circulation models and (re)analysis data is not sufficient to fully capture the small-scale aspects of such events related to boundary-layer dynamics. For instance, Škerlak et al. (2014) identified locations where a deep STT trajectory enters the boundary layer based on the 6-hourly diagnosed boundary layer height from ERA-Interim reanalyses. This approach cannot fully represent the sometimes highly dynamic diurnal cycle of the boundary layer height over continents. As an alternative,

Lefohn et al. (2011, 2012), used a subjective criterion for selecting deep STT trajectories that potentially affect the surface by requesting that the potential temperature of the trajectory is less than 5 K warmer than surface potential temperature. Their rationale was that a small vertical gradient of potential temperature can be overcome by boundary layer turbulence. Although such an assumption is reasonable, the choice of the 5 K threshold is arbitrary and with this approach it is neither possible to quantify the turbulent transport down to the surface nor to study the involved processes.

A promising methodological alternative to study the small-scale processes affecting deep STT in detail is high-resolution numerical modeling, for instance using weather prediction models with passive tracers of stratospheric air or chemistry transport models. However, also with this approach, several challenges occur: (i) high horizontal resolution is essential for well resolving complex topography and potentially the occurrence of convection (see type 4 in Fig. 1); (ii) at the same time the model domain should be large enough to include all processes from tropopause folding and isentropic transport to boundary layer mixing,

which can become prohibitively costly at high resolution; and (iii) numerical models, in particular if run at coarser resolution, suffer from numerical diffusion that can lead to spuriously strong downward transport of stratospheric air (e.g., Meloen et al., 2003). The idea to use passive tracers to quantify STT is not new: it has been successfully applied in a series of studies quantifying STT and identifying the involved physical processes (e.g., Kowol-Santen and Ancellet, 2000; Gray, 2003, 2006).

In this study we use a high-resolution regional numerical weather prediction model with an idealised stratospheric tracer

that is transported by resolved and sub-grid scale processes. In terms of model domain and resolution a compromise is made and the simulations are run with 7 km grid spacing in domains extending over more than 3000 km in both horizontal directions. Although we cannot hope to perfectly reproduce observed signals of stratospheric influence at the surface with this setup, the method allows us to directly quantify dilution by turbulent mixing along the whole transport pathway. The signature of the tracer observed at the lowest model level indicates whether a stratospheric influence at the surface (e.g., a peak in ozone) is

likely or not. However, we do not aim at estimating the associated increase of surface ozone [as done, e.g., by Wang et al. (2012) and Hofmann et al. (2016)], and therefore no direct comparisons will be made of our idealized tracer with surface observations. According to the ERA-Interim climatology by Škerlak et al. (2014), western North America and the Tibetan Plateau are global hotspots of deep STT [see also Lefohn et al. (2012) for North America and Lin et al. (2016) for the Tibetan Plateau]. From time series of surface ozone measured at Global Atmosphere Watch stations (GAW, e.g., Klausen et al., 2003),



we identified periods that were likely affected by intense STT events. For each of the two regions, we here present one such event that allows us to exemplarily highlight characteristic transport mechanisms.

The specific objectives of this numerical process study are: (1) to develop a complementary set of diagnostics that enables a quantitative investigation of stratospheric tracer transport by isentropic advection and turbulent mixing; (2) to better understand

the role of boundary layer turbulence in transporting stratospheric air down to the surface; and (3) to obtain a rough estimate of maximum surface concentrations of stratospheric tracer in the two case studies. In section 2 we describe the model setup. The case studies in western North America and on the Tibetan Plateau are presented in sections 3 and 4, respectively. Finally, our results are discussed in section 5 and the conclusions presented in section 6.

## 2   Data and Methodology

The simulations in this study are performed with the non-hydrostatic numerical weather prediction model COSMO (Baldauf et al., 2011), which is used operationally, among others, by the Swiss and German national weather services. We run COSMO with $0.0625°$ (approx. 7 km) horizontal resolution on a rotated geographical grid and 79 vertical hybrid levels up to approximately 15 hPa. The spacing of the vertical levels is smaller than 100 m in the lowest 5000 m a.s.l. over low topography and even up to 10'000 m a.s.l. over high topography. Our horizontal resolution is fairly high for STE case studies [e.g., Lin et al. (2012) used

a 50 km horizontal grid spacing] and allows for a detailed representation of small-scale flow features. This becomes especially important in steep mountainous regions, i.e., the target regions of this study (Fig. 2).

In the COSMO boundary layer scheme (Buzzi et al., 2011), a 1.5 order turbulence closure is applied where diffusion coefficients for vertical turbulent fluxes are computed using a parameterization based on turbulent kinetic energy. Convective up- and downdrafts as well as lateral convective entrainment are parameterised following Tiedtke (1989). At the boundary

of the COSMO domain, the meteorological fields are nudged towards operational IFS analyses from the ECMWF, which are interpolated in time (between 6-hourly fields) and space (from $0.5°$ horizontal resolution and 91 vertical levels).

Using a 3D labelling algorithm (Škerlak et al., 2014), we identify the dynamical tropopause (2-pvu isosurface in the extratropics and 380-K isentrope in the tropics) at every time instance of the IFS analyses and create a passive tracer field with the value 1 in the stratosphere and 0 in the troposphere. This tracer is handed over to the COSMO tracer advection scheme

(Roches and Fuhrer, 2012) and relaxed to the value of the IFS tracer at the lateral boundaries. There are no sources or sinks of this stratospheric tracer in the troposphere such that the tracer is transported in a fully passive way. Due to the lateral boundary conditions, the stratospheric tracer can only be transported across the tropopause and within the troposphere inside the COSMO domain. In addition to advection by the resolved wind fields, the tracer is transported by the parameterizations of sub-grid scale turbulence and convection.

The potential downward transport of the stratospheric tracer within the model domain requires a few hours up to some days. Therefore 7-day simulations are performed, initialized at 00 UTC 29 April 2006 for the event over North America, and at 00 UTC 9 June 2006 for the event over the Tibetan Plateau. As will be discussed below, in both cases first signals of the stratospheric tracer reached the surface after 1-2 days, and strong peaks occurred after 3-5 days.


**Figure 2.** Height (m a.s.l.) of orography in COSMO and domain of the simulations (orange boundaries) for the case studies (a) over western North America and (b) on the Tibetan Plateau. The red diamond in (a) indicates the location of Yellowstone National Park (YEL, 110.4°W, 44.6°N, 2468 m a.s.l.). The black frames denote 'target regions' used in the tracer evaluation (see below).





An important aspect of this study is the turbulent transport within the PBL and it is thus briefly described how the PBL top is identified from COSMO output. A widely used and robust method to determine the PBL top (Seibert et al., 2000) involves the bulk Richardson number, which is given by

$$Ri_B = \frac{g}{\bar{\theta}} \cdot \frac{\Delta\theta \cdot \Delta z}{(\Delta u)^2 + (\Delta v)^2}$$

where $\bar{\theta}$ is the average potential temperature in a layer of thickness $\Delta z$ (here the distance between two model levels) and $(u, v)$ are the horizontal wind components. The quantities $\Delta\theta$, $\Delta u$ and $\Delta v$ denote the differences of the respective values between the bottom and the top of this layer. The top of the PBL can then be defined as the height at which $Ri_B$ reaches a critical threshold. Theoretical and observational studies suggest a threshold of 0.25 (Nieuwstadt, 1984). In COSMO, the threshold is set to 0.33 in case of stable conditions, following Wetzel (1982), and 0.22 is chosen if the stratification near the surface is unstable or neutral

(Vogelzang and Holtslag, 1996). Which of the two cases is present is determined by the lapse rate of potential temperature in the lowest model levels. All PBL heights shown in this study are computed using this method.

## 3   Case Study I: Deep STT over North America

The selection of this event is also motivated by the observation of hourly averaged surface ozone concentrations of up to 89 ppbv at the remote mountain site Yellowstone National Park (label YEL in Fig. 2a) on 2 May 2006 (Lefohn et al., 2011). This station

is located on the eastern side of the Rocky Mountains. Lefohn et al. (2011), using STT trajectories and an empirical vertical stability criterion, estimated a high potential for turbulent downward mixing of an STT air mass to the surface. However, quantifying the dilution of the originally stratospheric air within the turbulent PBL had not been possible with this trajectory-based method.

### 3.1   Synoptic situation and formation of a tropopause fold

The main synoptic-scale flow feature of interest in this case study is an upper-level trough developing west of Alaska on 1 May 2006 as a positive PV anomaly on the 320-K isentrope (not shown). One day later, on 2 May, the trough has propagated over the continent. It has become narrow and its NW-SE oriented axis is aligned with the northern Rocky Mountains (Fig. 3a). On its western side, near 120°W, in the left entrance region of a jet streak, a tropopause fold has formed down to about 650 hPa, as can be seen in a meridional crosss section at 120°W (Fig. 3b). This corresponds well to the classical situation of a fold developing

within an amplifying Rossby wave (Keyser and Shapiro, 1986, their Fig. 19b). During the day, the upper-level frontal zone becomes more elongated, develops a clear cyclonic curvature, and extends to the east of the Rocky Mountains (Figs. 3c,e). This large-scale evolution in the model simulation compares well with ERA-Interim reanalyses (not shown).

### 3.2   Quasi-stationary front and transport to the surface

Due to the strongly tilted isentropes in the frontal zone associated with the tropopause fold (see Figs. 3b,d,f), air parcels can

be transported from the stratosphere to lower levels quasi isentropically. In fact, the frontal zone extends from upper levels

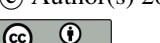



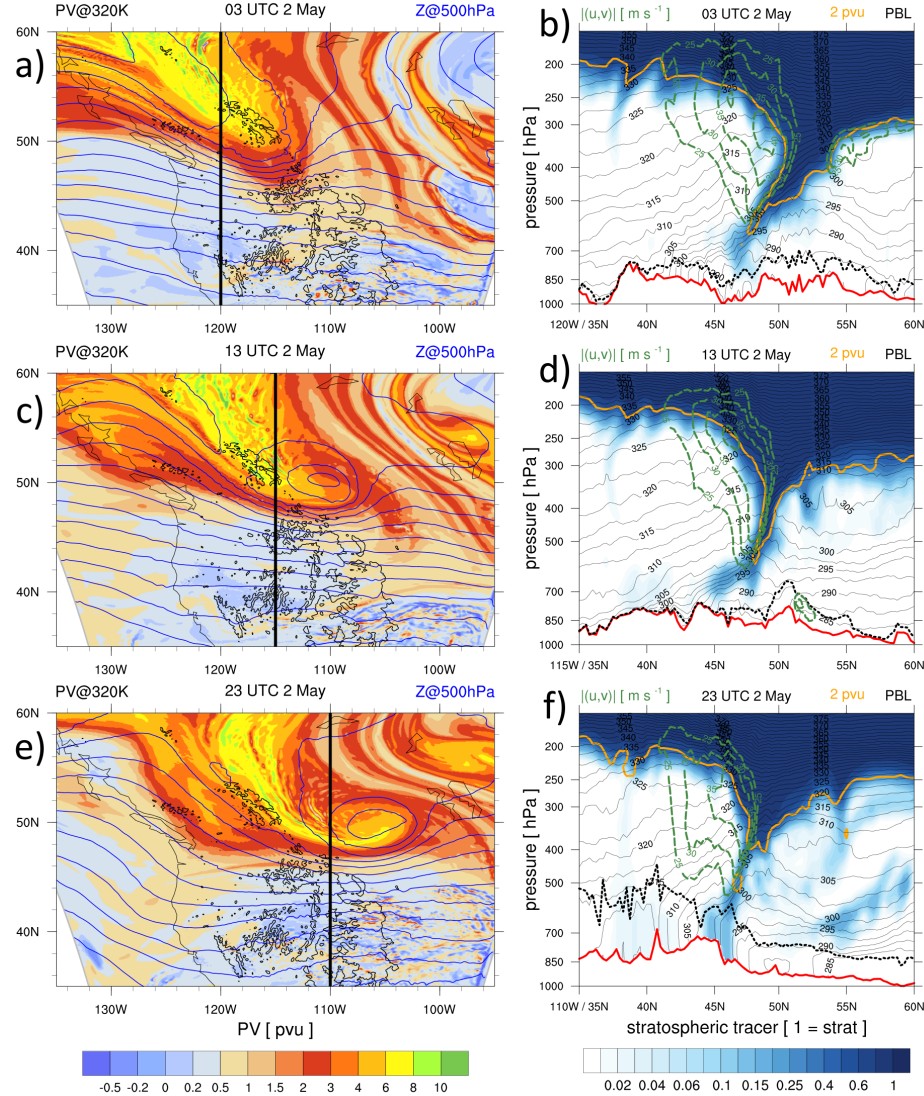

**Figure 3.** Left panels: PV on 320 K (in pvu, coloured) and geopotential height at 500 hPa (blue contours, interval 250 m) at (a) 03 UTC 2 May, (c) 13 UTC 2 May, (e) 23 UTC 2 May 2006. Right panels: Corresponding vertical sections across the tropopause fold from 35°N to 60°N (see black line in left panels). Depicted are concentration of stratospheric tracer (blue colours; a value of 1 denotes purely stratospheric air, note the non-linear scale), horizontal wind speed (in m s$^{-1}$, green dashed contours), 2-pvu isoline (orange contour), isentropes (in K, thin black contours), PBL height (dashed black contour), and topography (thick red contour).

(around 350 hPa) to the near surface (around 800 hPa), i.e., the upper-level and the near-surface cold fronts are connected. The strong frontal zone in the lower troposphere is visible from the potential temperature distribution at 700 hPa (not shown). The





zonally aligned front (roughly parallel to the zonally oriented tropopause in Fig. 3e) is not particularly strong (with a horizontal gradient of about $2\,\mathrm{K}\,(100\,\mathrm{km})^{-1}$) but remains quasi-stationary for nearly 48 h.

A significant amount of stratospheric tracer is found below the 2-pvu contour in Figs. 3b,d,f, which indicates that STT has occurred. This is especially the case beneath the jet stream in the lower parts of the frontal zone ($500-700\,\mathrm{hPa}$), where

turbulence typically is most intense due to instabilities triggered by strong vertical wind shear (Shapiro, 1980). The stratospheric tracer signal in the middle and low troposphere on 2 May 2006 is shown in Figs. 4a,c,e at times 03, 13 and 23 UTC. In the middle troposphere, i.e., on the 300-K isentrope, a band of high tracer concentrations elongates from the northern Rockies and rolls up cyclonically.

The ageostrophic response (secondary circulation) to the geostrophic forcing near the front results in rising motions in a

comparatively narrow band ahead of the cold front and subsidence over larger areas behind it (see, e.g., Sawyer, 1956). In our simulation, we observe varying mesoscale patterns in the vertical wind fields near the front (not shown). Hence, locally, the frontal ageostrophic circulation most likely brings stratospheric tracer from the free troposphere (around $700\,\mathrm{hPa}$) down to the surface after about 13 UTC 2 May 2006, when the tracer covers a large fraction of the target region on 300 K (Fig. 4c). But the most important transport mechanism remains the growth of the PBL during daytime. Indeed, in the afternoon of 2 May

2006, the PBL top reaches levels of roughly $600\,\mathrm{hPa}$ as can be seen from the vertical cross-section in Fig. 3f – keeping in mind that local time is UTC$-6\,\mathrm{h}$. The stratospheric tracer is entrained at the PBL top and transported to the surface by turbulent motions. Therefore, high concentrations up to 20% of the stratopsheric tracer occur at near-surface levels in the second half of 2 May (Fig. 4f). In agreement with substantial contributions from frontal downward transport, an almost linear and very narrow structure of strongly enhanced tracer concentrations emerges at the surface.

Ozone concentrations measured at YEL rapidly increased in the afternoon of 2 May 2006, reaching 89 ppbv at 19 UTC (Lefohn et al., 2011). The timing of this increase agrees well with the time series of the stratospheric tracer on the lowest model level interpolated to YEL (not shown). Assuming an ozone concentration of 200 ppbv in the lowermost stratosphere and taking into account that tracer values at the surface reach about 20%, deep STT could largely explain the observed increase of 40 ppbv during 2 May 2006. Unfortanely, no further ozone observations are available to also qualitatively assess the spatial

structure of the simulated STT event down to the surface.

## 4 Case Study II: Deep STT over the Tibetan Plateau

The Tibetan Plateau is the world's largest and highest plateau with the average altitude exceeding 4500 m a.s.l. (see Fig. 2b). It is bounded by the Himalayas in the south, the Pamirs in the west and the Kunlun Mountains in the North. Intense diurnal heating over the Tibetan Plateau can create a deep layer of large-scale ascent and enables vigorous near-surface turbulence,

especially in its semi-arid western part (Yanai and Li, 1994). Combined with the already very high orography, these conditions allow for extreme PBL heights reaching up to 9500 m a.s.l. (Chen et al., 2013, 2016).

Although climatologically spring is the peak season for deep STT in this region (Škerlak, 2014; Škerlak et al., 2014), here we study an STT event in summer 2006 when we observed a prominent tropopause fold in ERA-Interim reanalyses that approached



**Figure 4.** Concentration of stratospheric tracer (blue colors up to values of 0.2; green shows values larger than 0.5) on the 300-K isentrope (left) at (a) 03 UTC 2 May, (c) 13 UTC 2 May and (e) 23 UTC 2 May 2006 (as in Fig. 3) and (right) on the lowest model level at the same times. Additionally, the geopotential height at 500 hPa is shown with orange contours (contour interval 250 m) in the right panels.

the Tibetan Plateau from the west. The simulation results (see below) then strongly supported the assumption that this episode could have led to STT down to the surface. Unfortunately, no representative surface station data from the Plateau is available



to validate this inference. Therefore, we focus in the following on identifying and understanding the processes leading to the surface signals of the stratospheric tracer in the model simulation.

## 4.1 Synoptic situation and formation of a tropopause fold

As for the case study over North America, the main synoptic feature of interest is an upper-level frontal zone, which is characterized by a region of intense horizontal gradients of potential temperature on 200 hPa (not shown). It is associated with a cyclonic PV anomaly that developed over the Kasakh Steppe during the previous few days (Fig. 5a). At 00 UTC 13 June, the strong upper-level front is situated at the eastern side of the Pamir mountains. The jet stream is strongly cyclonically curved with cold and warm air on its cyclonic and anticyclonic shear side, respectively [corresponding to the 'thermal trough' situation according to Keyser and Shapiro (1986, their Fig. 23f)].

Figure 5b shows the vertical structure of this upper-level front at 00 UTC 13 June in a cross-section along the black line in Fig. 5a. Neither wind speed maxima (around $30\,\mathrm{m\,s^{-1}}$) nor the depth of the tropopause fold (down to 400 hPa) are unusual. Nevertheless, STT evidently has occurred since the concentrations of the stratospheric tracer are already enhanced below the 2-pvu contour. In stark contrast to the previous case study (cf. Fig. 3), no baroclinic zone is present at lower levels and the frontal zone is thus confined to the upper troposphere. Additional vertical cross-sections are shown in Fig. 5d,f at 00 and 09 UTC 14 June, but aligned in the along-flow direction (see black lines Figs. 5c,e). A layer with high concentrations of the stratospheric tracer is discernible between 300 and 500 hPa at 00 UTC (Fig. 5d), extending from 75°E to the Tibetan Plateau at 90°E. The top of this layer, near 300 hPa and 350 K, is still characterized by stratospheric PV values. The three vertical cross-sections already indicate that ozone surface signals over this target region are determined by several processes: STT within the tropopause fold, the subsequent quasi-horizontal transport over the Tibetan Plateau, and (potentially) a strong impact of the PBL's daily cycle [note the deep well-mixed PBL over the Plateau in the afternoon of 14 June (Fig. 5f)]. In the following, we will further elaborate on this threefold chain of processes.

## 4.2 Advection over the Tibetan Plateau and transport to the surface

As discussed before, the increased concentrations of the stratospheric tracer below the 2-pvu surface indicate that STT evidently has occurred in the upstream tropopause fold (Fig. 5b). Within the tropopause fold and after crossing the tropopause, these air masses descend and spread out on the slanting isentropes down to levels of approximately $300-500$ hPa. The potential temperature of the stratospheric air parcels range from $330-350$ K within the tropopause fold and reach values of about 325 K in its immediate neighborhood. During this phase, mixing with surrounding tropospheric air masses reduces the concentrations of the stratospheric tracer, but comparatively high values of up to 60% are found in the surrounding of the fold.

The vertical cross-section in Fig. 5d at 00 UTC 14 June and oriented along the flow indicates that the originally stratospheric air masses are then quasi-horizontally transported from the tropopause fold across the Tibetan Plateau. This transport occurs at a fairly constant pressure level between $300-500$ hPa and is quasi-isentropic in the layer between $325-340$ K. Further, this layer of intruding stratospheric air between $300-500$ hPa is clearly separated from the stratospheric reservoir above 150 hPa. The transport of the stratospheric tracer at the top of the intrusion band at 340 K is depicted in Fig. 6a,c,e. The overall structure



**Figure 5.** Left panels: PV on 350 K (in pvu, coloured) and geopotential height at 500 hPa (blue contours, interval 250 m) at (a) 00 UTC 13 June, (c) 00 UTC 14 June, (e) 09 UTC 14 June 2014. Right panels: Corresponding vertical sections across the tropopause fold as indicated by the black lines in the left panels. Depicted are concentration of stratospheric tracer (blue colours; a value of 1 denotes purely stratospheric air, note the non-linear scale), horizontal wind speed (in m s$^{-1}$, green dashed contours), 2-pvu isoline (orange contour), isentropes (in K, thin black contours), PBL height (dashed black contour), and topography (thick red contour).







**Figure 6.** Concentration of stratospheric tracer (blue colors up to values of 0.2; green shows values larger than 0.5) on the 340-K isentrope (left) at (a) 00 UTC 13 June, (c) 00 UTC 14 June and (e) 09 UTC 14 June 2006 (as in Fig. 5) and (right) on the lowest model level at the same times. Additionally, the geopotential height at 500 hPa is shown with orange contours (contour interval 250 m) in the right panels.



with cyclonically wrapping tongues of dry stratospheric air and moister mainly tropospheric air over the Plateau and further north agrees qualitatively well with water vapour satellite images (not shown). At 09 UTC 14 June, the concentration of the stratospheric tracer on 340 K near the center of the target region (the polygon in the figure) is still rather high, reaching values larger than 0.5 (see green colours), i.e., indicating only weakly diluted stratospheric air.

In the early morning of 14 June 2006, at 00 UTC (local time is UTC + 7), the PBL is very shallow (Fig. 5d) and virtually no stratospheric tracer is found at the surface (Fig. 6d). During the next 9 h, however, the PBL grows steadily and extends vertically up to 300 hPa (Fig. 5f). As a result, entrainment at the top of the growing PBL and turbulent mixing within the PBL transport stratospheric tracer from the upper troposphere down to the surface. This leads to a strong tracer signal on the lowest model level at 09 UTC 14 June (Fig. 6f) with peak values of 20%. In terms of potential temperature, the PBL over the
Tibetan Plateau is characterized by nearly uniform values around 330-340 K, i.e., the air masses above the Plateau are well mixed due to turbulence in the PBL up to 340 K (corresponding to about 350 hPa). This is remarkable, as it indicates that the PBL can grow vertically up to the isentropic level on which the large-scale advection of stratospheric tracer from the upstream tropopause fold occurred previously. The turbulent downward transport from the top of the PBL at 340 K is also discernible in the time evolution of the stratospheric tracer concentation at 340 K: Whereas the tracer concentrations reached value larger
than 0.5 in large parts of the target region at 00 UTC 14 June (green colours in Fig. 6c), the values are significantly reduced 9 h later (Fig. 6e). In agreement with the onset of PBL mixing, the surface concentrations of the stratospheric tracer subtantially increase during these 9 hours (Figs. 6d,f).

## 5   Discussion and a refined analysis

The processes leading to the transport of stratospheric air down to the surface in the first case study (Rocky Mountains) can
be classified as a combination of types 2 and 3 as defined by Johnson and Viezee (1967) (see Fig. 1), i.e., both secondary circulations near a surface front and turbulent mixing within the PBL are likely to contribute to the transport of the tracer down to the surface. Further, this case study fits nicely with the concept of a strongly subsiding stratospheric intrusion (e.g., Danielsen, 1964; Browning, 1997) behind the cold front of an extratropical cyclone. As a side remark, it is noted that the strongest descent in the rear of an extratropical cyclone, the so-called 'dry intrusion', is typically of tropospheric origin (Wernli,
1997; Raveh-Rubin, 2017).

In the second case study (Tibetan Plateau) the extreme PBL heights clearly indicate a variant of type 2 according to the classification by Johnson and Viezee (1967). But, in contrast to the schematic illustration in Fig. 1, no significant vertical transport in the free troposphere was required in our case. Indeed, the tropopause fold merely brings stratospheric air to levels between $300 - 500$ hPa. No connection to a surface baroclinic zone is apparent in this case. The crucial phase of large-scale
transport is the advection of originally stratospheric air in the layer above 500 hPa across the Tibetan Plateau, where it is entrained into the growing PBL and mixed down to the surface. As a consequence, the tracer signature at the surface at 09 UTC 14 June looks similar to the one at 340 K (compare Figs. 6e,f): PBL turbulence 'imprints' the structure established first by large-scale advecction in the mid-troposphere to the surface. This case study provides interesting insight into the processes of type 2



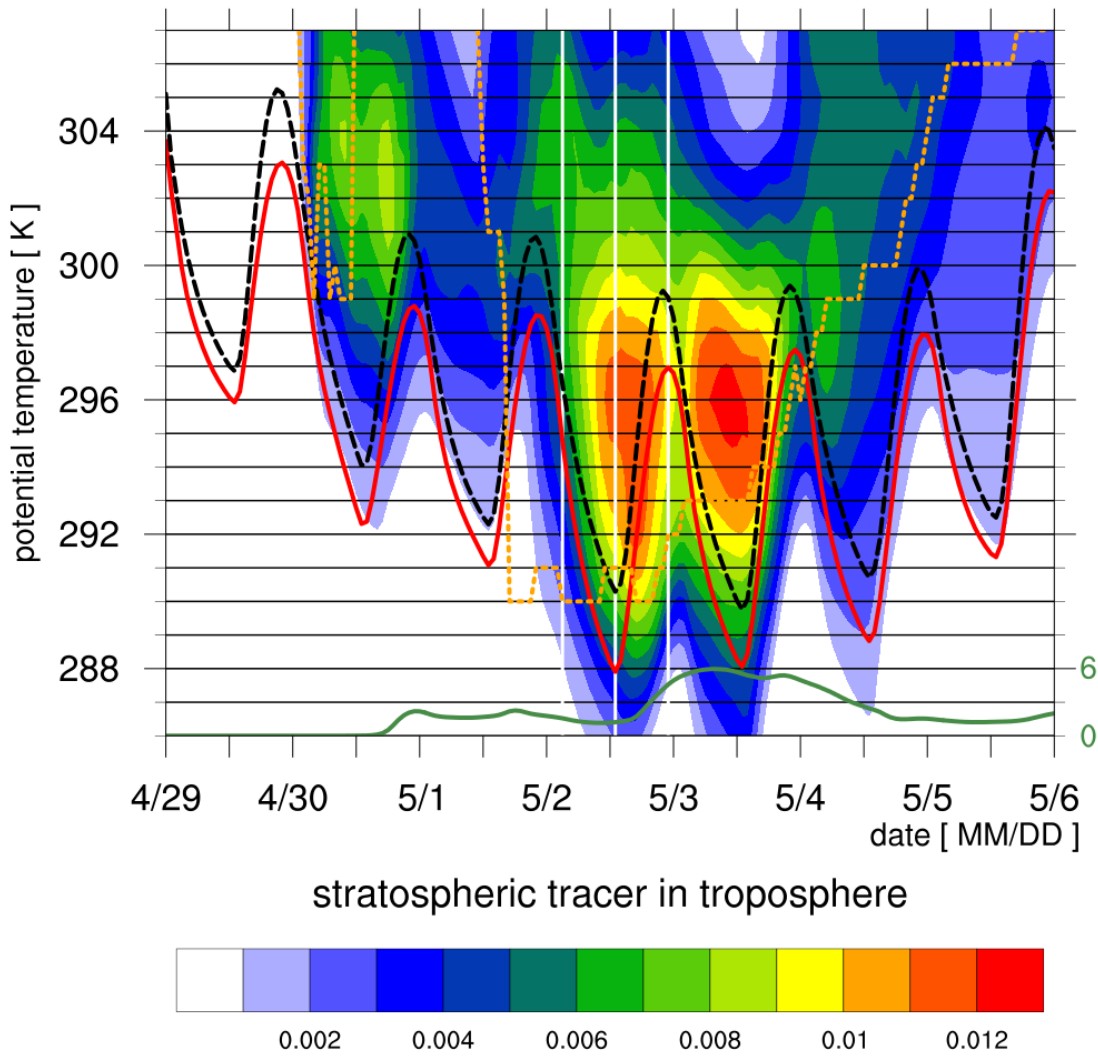

stratospheric tracer in troposphere

**Figure 7.** Time series for the North American STT event from 29 April to 06 May 2006 of the stratospheric tracer concentration (colours; averaged over all tropospheric grid points in the target region polygon shown in Fig. 2a on several isentropic surfaces), the surface height (solid red line, averaged over target region), the height of the top of the boundary layer (dashed black line, averaged also over the target region), and the lowest potential temperature value of the tropopause in the target region (dashed orange line). The green line indicates the averaged stratospheric tracer concentration in the target region on the lowest model level. The value of 6 corresponds to 0.6%. The white vertical lines denote the times shown in Figs. 3 and 4.

events. It shows that, even in summer, deep STT can impact surface air over the Tibetan Plateau by means of quasi-horizontal transport followed by entrainment into the extremely high-reaching PBL.



**Figure 8.** Time series as in Fig. 7 but for the Tibetan Plateau STT event from 09 to 16 June 2006. The white vertical lines denote the times shown in Figs. 5 and 6.

The different nature of STT down to the surface in the two case studies can be further elucidated with the type of diagrams shown in Figs. 7 and 8. These figures show time series of several relevant parameters in the target region polygons marked in Figs. 2, 4 and 6. Colors represent the stratospheric tracer concentration averaged over tropospheric grid points on isentropes from 286 to 307 K for the North American case (Fig. 7), and from 320 to 350 K for the Tibetan Plateau case (Fig. 8). Once the

5  stratospheric tracer has entered the troposphere (via STT), the tracer spreads horizontally in these diagrams due to isentropic





advection. In contrast, a vertical expansion of the stratospheric plume towards lower isentropes is only possible if another STT event occurs in the model domain on the lower isentrope, or via turbulent mixing. Red and black lines show the time evolution of the averaged potential temperature value of the surface and the top of the PBL, respectively. Due to the typically weak stratification in the well-mixed PBL, the two values are similar and evolve in parallel. The much more fluctuating orange line then indicates the isentropic level of the lowest point of the tropopause within the target polygon. Ott et al. (2016) showed similar diagrams (their Figs. 6 and 10) for analyzing summertime STT events over Maryland, U.S., but using height instead of potential temperature as the vertical coordinate, which makes the separation of isentropic advection and cross-isentropic turbulent mixing less obvious.

The usefulness of the $\theta - t$ tracer concentration diagrams and combination of parameters becomes apparent when considering the evolution after 00 UTC 30 April in Fig. 7. The steep decrease of the orange line at this time to lower potential temperature values indicates that a first stratospheric intrusion (not yet the one shown on 02 May in Fig. 3a) enters the target region[1]. Now as the intrusion reaches down to 299 K, STT occurs along the edge of this intrusion as indicated by the non-zero values of the stratospheric tracer concentration in the troposphere. Elevated tracer concentrations (green values) only occur on isentropes $\geq 299$ K due to STT; the weaker (blueish) values that reach down to 292 K during the next 12 hours are a consequence of turbulent entrainment into the PBL. Until 12 UTC 01 May, some of the stratospheric tracer is advected out of the target polygon, as indicated by the decreasing mean concentration values.

The deep trough shown in Fig. 3a and in Fig. 4a enters the target region after 12 UTC 01 May, as shown again by the steep orange line. In this case, the lowest point of the tropopause remains as low as 290 K until 00 UTC 03 May, such that intense STT occurring during these 36 hours between $290 - 310$ K leads to high concentrations of the stratospheric tracer in this fairly thick isentropic layer. Recall that Figs. 4a,c,e showed the time evolution of the stratospheric tracer exactly in the middle of this layer on 300 K. The first peak of averaged tracer concentrations occurs at about 12 UTC 02 May on 296 K. During the next 6-12 hours, due to the diurnal growth of the PBL up to 297 K, the stratospheric tracer can be transported isentropically down to the surface. This is exactly the process described above, with the steep isentropes between $295 - 305$ K intersecting the surface (see Figs. 3d,f). The green line in the bottom of the diagram shows the concentration of the tracer on the lowest model level, horizontally averaged in the target polygon (see Fig. 2b), which increases strongly during this episode.

The same type of diagram for the STT event on the Tibetan Plateau (Fig. 8) shows several common features, but also some relevant physical differences. On the first two days, the target region was unaffected by STT. Then the deep trough mentioned above (Fig. 5a) entered the target region on 11 June on isentropes down to 326 K and some stratospheric tracer already appears near the surface (see green line at the bottom of the diagram). During the next days, intense STT and horizontal advection occurs mainly in the layer between 335-340 K, indicated by the strong increase with time of the stratospheric tracer in the target region on these isentropes. As discussed in the previous section, the deeply growing PBL over the Tibetan Plateau then grows up to 336 K, i.e., it reaches into the layer enriched with stratospheric tracer, which is then transported efficiently down to surface, in particular in the evenings of 14 and 15 June (in agreement with Fig. 6f). An important difference to the North

---

[1] The fact that the orange line touches the black and red line does not indicate that the tropopause touches the ground; note that the black and red lines show spatially averaged values, whereas the orange line represents a single grid point.





American case is that here the maxima of the stratospheric tracer concentration occur on isentropes that are slightly higher than the maximum height of the boundary layer. This is consistent with the fact that isentropic transport down to surface is not possible in this case; it can only occur due to turbulence in the PBL characterized by a strong diurnal cycle.

## 6 Conclusions

Two warm season case studies have been presented to study the processes leading to the transport of originally stratospheric air down to the surface in regions with complex topography. To this end, mesoscale model simulations were performed with an idealized passive tracer initialized in the stratosphere. For an STT event over the Rocky Mountains and another one over the Tibetan Plateau, the evolution of this tracer was followed from the tropopause fold, where the tracer entered into the troposphere, along the long-range isentropic transport to the turbulent mixing in the planetary boundary layer. The main conclusions of this model-based process study can be summarized as follows:

– The approach allows for quantitatively differentiating between isentropic transport and vertical, i.e., cross-isentropic transport and dilution due to turbulent mixing.

– The combination of complementary diagnostics (tracer evolution on isentropic surfaces, vertical cross sections and a novel diagram of horizontally averaged tracer concentrations as a function of potential temperature and time) led to detailed insight into the main processes responsible for the downward transport of stratospheric tracer.

– Processes involved in two contrasting types of deep STT have been portrayed: In situations with a troposphere-deep frontal zone (North America case study), isentropic descent can directly transport stratospheric tracer to near-surface layers where turbulent mixing occurs quasi-simultaneously. In contrast, in regions with vertically confined upper-level fronts (Tibetan Plateau case study), quasi-horizontal isentropic advection in the mid-troposphere is mainly responsible for the transport from the tropopause fold to the high-elevation plateau, where the stratospheric tracer is entrained from below by the very deeply growing boundary layer. Here the downward transport of the tracer by turbulent mixing is decoupled from the tropopause fold that brought stratospheric air into the troposphere in the first place.

– For both STT events, despite their differing meteorological setting, peak surface concentrations of the stratospheric tracer reached values of about 20%. It is thus plausible that such events can lead to enhancements of surface ozone concentration by up to 50 ppbv.

– The results emphasize the multi-scale nature of STT, in particular if interested in potential effects of STT at the surface. The involved isentropic transport can extend over several 1000 km, whereas the eventual turbulent downward transport is affected by local topography and boundary layer dynamics. The imposed need for high-resolution modeling clearly illustrates the challenge in simulating surface effects of STT globally and over climatological time periods.

There are three main limitations of this study, related to (i) the low number of investigated STT events, (ii) the neglectance of tropospheric chemistry, and (iii) the missing link to surface observations. Chemistry has been neglected on purpose, in order to



focus on isentropic and turbulent transport. Finding a representative set of surface observations is challenging and we therefore focused entirely on identifying and understanding processes as simluated by a state-of-the-art mesoscale model. And finally, we acknowledge that two case studies are of course not representative. But they serve well to illustrate the important case-to-case variability when considering processes leading to STT down to the surface, and to emphasize the key role of the larger-scale

5 setting in which STT occurs (deep front vs. upper-level front; interaction of front with boundary layer dynamics).

*Data availability.* Output of the numerical simulations is available from the authors upon request.

*Author contributions.* All authors designed this study; BS performed and evaluated the numerical simulations and wrote a first draft of the paper; all authors discussed the results and contributed to the final writing of the paper.

*Competing interests.* The authors declare no competing interests.

10 *Acknowledgements.* We are grateful for the support of Anne Roches with implementing the stratospheric tracer in COSMO.



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
