# Peer review of "A numerical process study on the rapid transport of stratospheric air down to the surface over western North America and the Tibetan Plateau"

_Atmospheric Chemistry and Physics, 2018_

## Referee Comment (RC1) · Gray (Referee) · 7 Jan 2019

General comments:

This is a very nice paper that presents a relatively high resolution modelling study (7 km horizontal grid spacing) of two cases of stratosphere to troposphere transport. In both cases stratospheric air descends down to the planetary boundary layer though the mechanisms leading to that descent differ for the two cases. The paper is clearly and concisely written and the results support the conclusions drawn. I recommend it for publishing subject to consideration of the following minor comments.

[Figure]

Specific comments:

1. Abstract p1 L16: Here it says that that the surface concentrations reach peak values of 10-20% in both cases which raises the question '% of what'. Add 'of the imposed stratospheric value' or similar.

2. Although the focus of the study is on the processes leading to the transport of air from the stratosphere down to the boundary layer, rather than on the concentrations of tracer reached in the boundary layer, the paper would benefit from comparison of the concentrations found with those found in some of the comparable (i.e. using passive tracers) modelling studies referenced in the introduction to the paper. The papers referenced don't focus on transport down to the boundary layer, but it might be useful to compare the transport into the upper-troposphere with your study since this transport is an essential pre-requisite to the deeper transport.

3. Robustness of transport across tropopause: Related to the above comment, it would be useful to include some discussion as to the potential issues with the realism of the transport across the tropopause surface as this is likely to be strongly dependent on the vertical resolution and on the implicit (and any explicit) diffusion in the model (which may or may not be realistic). Presumably the tracer is initialised as a step function across the tropopause - the model would be unlikely to be able to retain this step function even in the absence of any 'physically realistic' transport. How does the tracer advection scheme differ from the advection scheme used for prognostic variables in the model?

4. Importance of convection: how important is convection in your case studies? The daytime growth of the boundary layer and associated turbulent mixing is key to transport in both cases, but there is no mention of the possible role of convective mixing (the type 4 mechanism in your Fig. 1). If convection does have a role in your case studies you might want to consider whether a convection-permitting simulation may yield more or less transport into the boundary layer. For example, we (Chagnon and

Gray, 2010: https://doi.org/10.1029/2010JD014421) found relative insensitivity to the explicit representation of convection. There was a reduction in the spatial extent in the tracer transported into the lower-troposphere. In one of the cases, a case with mid-level convection, there was a substantial increase in the total tracer transported to lower tropospheric levels with explicit convection (but very little difference for the other two cases).

Technical corrections:

1. p1 L21: change to 'studies have revealed'.

2. p2 L10: change to 'question of whether'.

3. p5, L11: The resolution will be several times (~6) that of the grid spacing which is the value you give here.

4. Fig 3-6: Labelling of geopotential height contours. I appreciate that the contour interval is included in the caption, but there is no indication of actual values. Can at least a few of the contours be labelled please?

5. Fig 5, right column panels: It would be nice if the domains of the cross sections could be extended to the left so that the start region of the layer of high stratospheric tracer concentrations could be determined.

6. p11, L19: Please move the statement about the relationship of local and UTC time from p14 L4 to here (where the text refers to the 'afternoon of 14 June').

7. p14, L32: correct typo 'advecction'.

8. p18, L10: add 'transport' after 'vertical'.

9. p18. Consider hyphenating 'horizontally-averaged' and 'vertically-confined' to aid readability.

[Figure]

2018.

---

## Referee Comment (RC2) · Lefohn (Referee) · 13 Jan 2019

**Review**

**A numerical process study on the rapid transport of stratospheric air down to the surface over western North America and the Tibetan Plateau**

Bojan Škerlak, Stephan Pfahl, Michael Sprenger, and Heini Wernli

**General**

Ozone contributions originating from the stratosphere can affect surface ozone concentrations by presenting themselves as episodic events (i.e., high hourly averaged concentrations for a relatively short period of time) or as enhanced surface concentrations (i.e., moderate hourly average enhancements sometimes over a short time or over a longer time period). The authors have focused on investigating episodic ozone events which occurred at two different locations (i.e., Yellowstone National Park (NP) and the Tibetan Plateau) using a passive stratospheric air mass tracer in a mesoscale model to explore the processes that enable the transport down to the surface. The events occurred in early May 2006 at Yellowstone NP and in mid-June 2006 on the Tibetan Plateau. In both cases, a tropopause fold associated with an upper-level front enabled stratospheric air to enter the troposphere. Despite the strongly differing dynamical processes at the two locations, stratospheric tracer concentrations at the surface reached peak values of 10-20% (i.e., 20-40 ppb) that added to the surface concentrations, corroborating the potential of deep stratosphere-to-troposphere transport events to episodically influence surface ozone concentrations in these two regions.

The authors noted 3 limitations of their study were (1) the low number of investigated STT events, (2) the negligence of tropospheric chemistry, and (3) the missing link to surface observations. While the authors were able to compare their model predictions with the recorded hourly ozone data at Yellowstone NP, the authors indicated that no representative surface station data were available from the Tibetan Plateau. Therefore, while comparing their predictions with data from Yellowstone NP, they were unable to validate their predictions for the Tibetan Plateau. Recognizing this problem, they focused their attention on identifying and understanding the processes leading to the surface signals of the stratospheric tracer in the model simulation.

I do have two concerns that I believe the authors should address. My first concern deals with the statement that the number of "deep STT" events is rare. The the use of the term "deep STT" is used differently by researchers who have published in the literature. In the manuscript, the authors used the term "deep STT" as discussed in Škerlak et al. (2014). Škerlak et al. (2014) defined "deep STE" as stratospheric air that reaches the PBL *within 4 days or vice versa*. The definition of "deep STT" (or STE) in the manuscript focuses on the processes required to transport the stratospheric air down to the PBL over a very short period and then down to the surface. Škerlak et al. (2014) note that for a subset of deep STE events, the downward ozone flux into the PBL is dominated by the mass flux and are most frequent in early spring. The authors concluded that surface ozone concentration along the west coast of North America and around the Tibetan Plateau are likely to be influenced by deep stratospheric intrusions.

As noted by the authors, Lefohn et al. (2011 and 2012) described the influence of STT trajectories on surface ozone at Yellowstone NP. In their papers, Lefohn et al. (2011 and 2012) discuss the frequency of deep STT "hits" that are associated with periods greater than the 4-day criterion applied by Škerlak et al. (2014). Thus, I believe there is some confusion about the use of the term "deep STT". While the current manuscript refers to the rare occurrences of "deep STT", other papers indicate a greater number of occurrences of "deep STT" using a different set of criteria for the term "deep STT".

I am not necessarily convinced that the "deep STT" events are as rare as the authors indicate in their manuscript if a broader definition of the term "deep STT" is applied. It is obvious that every "deep STT" event will not necessarily lead to an episodic enhancement of ozone (e.g., 20-40 ppb) at the surface. For example, Lefohn et al. (2011 and 2012) quantified the number of STT-S "hits" that occurred at the Yellowstone NP site. Lefohn et al. (2011, 2012 used a subjective criterion for selecting deep STT trajectories that potentially affect the surface by requesting that the potential temperature of the trajectory is less than 5K warmer than surface potential temperature. Their rationale was that a small vertical gradient of potential temperature can be overcome by boundary layer turbulence. While Lefohn et al. (2011 and 2012) were unable to quantify the turbulent transport down to the surface or study the involved processes, they did quantify the number of STT trajectories (STT-S) that were predicted to reach the surface and potentially enhance surface ozone concentrations. Lefohn et al. (2011) noted for sites across the US the following:

1. For the time period of their analyses, the high-elevation site at Yellowstone National Park (NP) in Wyoming exhibited more than 19 days a month during the spring and summer for hourly average ozone concentrations ≥ 50 ppb with STT-S> 0, where STT-S represented the number of deep trajectories. At this site, the daily maximum hourly springtime average ozone concentrations were usually in the 60-70 ppb range. The maximum daily 8-h average concentrations mostly ranged from 50 to 65 ppb;

2. At many of the lower-elevation sites, there was a preference for ozone enhancements to be coincident with STT-S> 0 during the springtime, although summertime occurrences were sometimes observed; and

3. For many cases, the coincidences between the enhancements and the STT-S events occurred over a continuous multiday period. When statistically significant coincidences occurred, the daily maximum hourly average concentrations were mostly in the 50-65 ppb range and the daily maximum 8-h average concentrations were usually in the 50-62 ppb range;

Lefohn et al. (2012) also noted:

1. STT down to the surface (STT-S) frequently contributed to enhanced surface ozone hourly averaged concentrations at sites across the US, with substantial year-to-year variability;

2.  The ozone concentrations associated with the STT-S events appeared to be large enough to enhance the measured ozone concentrations during specific months of the year;

3.  Months with a statistically significant coincidence between enhanced ozone concentrations and STT-S occur most frequently at the high-elevation sites in the Intermountain West, as well as at the high-elevation sites in the West and East;

4.  These sites exhibit a preference for coincidences during the springtime and in some cases, the summer, fall, and late winter; and

5.  Besides the high-elevation monitoring sites, low-elevation monitoring sites across the entire US experienced enhanced ozone concentrations coincident with STT-S events.

As indicate above, the number of STT-S "hits" as described by Lefohn et al. (2011, 2012) were frequent at many sites. For example, for the Yellowstone NP site in 2006, the figure below illustrates the daily maximum 8-h observed concentration and the number of daily STT-S "hits" over the entire year.

[Figure]

While the number of STT-S "hits" were frequently found at high-elevation sites in the US during the spring and summer, they were also found at times during the springtime at low-elevation sites. The number of STT-S hits were mostly associated with subtle enhancements of surface

ozone concentrations when compared to the number of episodic events. In other words, many of the "deep STT" events were associated with subtle enhancements to surface ozone, while other less frequent "deep STT" events were associated with episodic additions to the surface levels. In summary, "deep STT" events contributed to the more frequent enhancements of surface ozone concentrations rather than the episodic events that raised surface ozone levels (20-40 ppb).

I would recommend that the authors carefully define in their manuscript the term "deep STT" and caution the reader that other researchers have reported deep STT events using different criteria than those used by the authors. Clearly, other published works have discussed the importance of stratospheric ozone transport into the PBL to shape the distribution of hourly average concentrations at both high- and low-elevation sites. For example, the Mt. Waliguan site on the Tibetan Plateau is highly representative of free-tropospheric ozone (Ma et al., 2002) and is often influenced by stratosphere-to-troposphere transport (STT) events (Ding and Wang, 2006; Zhu et al., 2004).

Perhaps the authors should expand their discussion in the Introduction to include their views on the importance of STT processes that influence surface ozone concentrations that include both episodic, as well as subtle enhancements. I believe a more balanced discussion should be considered.

My second concern is associated with the inability of the authors to validate their predictions with a site in the Tibetan Plateau. Hourly ozone data for Mt. Waliguan, China (Latitude 36° 17' N; Longitude 100° 54' E) have been recorded for June 2006. The authors might wish to obtain the hourly ozone data by requesting the information for this time period by contacting Dr. Xiaobin Xu (Key Laboratory for Atmospheric Chemistry, Institute of Atmospheric Composition, Chinese Academy of Meteorological Sciences, China Meteorological Administration, Zhongguancun Nandajie 46, Beijing 100081, China - xiaobin_xu@189.cn). On 17 June 2006 at 0300 Beijing Time (UTC + 8 hours), the hourly average ozone concentration was similar in magnitude to the episode that occurred at Yellowstone NP described in the manuscript. The site is situated at the northeastern edge of the Tibetan Plateau. While the Mt. Waliguan site is outside of the target region defined in the manuscript, the authors might wish to modify their analyses so that their predictions can be compared with actual ozone data recorded in the Tibetan Plateau for the June 2006 period. This is a decision I will leave to the authors and perhaps the editor.

I would recommend that the manuscript be published once my first concern is addressed. As indicated above, my second concern about the addition of the Mt. Waliguan data to their analyses is a decision left to the authors and perhaps the editor.

**Specific Suggestions**

*Abstract*

Page 1, Line 2-3: I suggest changing "these plumes of" to "mid-June plumes associated with".

Page 1, Line 4-6: I suggest changing "(ii) if boundary layer turbulence is strong enough to enable this transport, the originally stratospheric air mass is strongly diluted by mixing such that only a weak stratospheric signal can be recorded at the surface" to "(ii) even if boundary layer turbulence were strong enough to enable this transport, the originally stratospheric air mass can be diluted by mixing, such that only a weak stratospheric signal can be recorded at the surface."

Page 1, Line 6: I suggest changing "from" to "associated with."

Page 1, Lines 8-9: I suggest changing "The events occurred in early May 2006 in the Rocky Mountains and in mid June 2006 on the Tibetan Plateau" to "The events occurred in early May 2006 at Yellowstone National Park in Wyoming and in mid-June 2006 on the Tibetan Plateau."

Page 1, Line 12: I suggest changing "and a reservoir" to "and initially a reservoir."

Page 1, Line 13: I suggest changing "entrainment" to "However, entrainment…"

Page 1, Line 14: I suggest changing "fosters" to "allows for…"

Page 1, Line 15: I suggest removing the word "Interestingly" and starting the sentence with "Despite."

Page 1, Line 16: Would it be possible to place into parenthesis the range of ozone concentrations at the surface such as 10-20% (i.e., 20-40 ppb)" or whatever the range of concentrations is?

*Introduction*

Page 1, Line 19: I suggest replacing "known" with "documented."

Page 2, Line 1: I would suggest citing Langford et al. (2009). Langford, A.O., Aikin, K.C., Eubank, C.S., Williams, E.J., 2009. Stratospheric contribution to high surface ozone in Colorado during springtime. Geophysical Research Letters 36, L12801. http://dx.doi.org/10.1029/2009GL038367.

Page 2, Lines 4-5: I would suggest changing "cross the tropopause and then to descend down to" to "cross the tropopause and to descend to".

Page 2, Lines 5-6: I would suggest changing "This isentropic descent typically goes along with" to "This isentropic descent typically is accompanied with".

Page 2, Line 12: I would suggest changing "that deep STT events are comparatively rare" to "that deep STT events are comparatively infrequent…" (Again, I would suggest that the authors re-evaluate how "rare "deep STT" events are. Several authors have reported frequent "deep STT" events that are not necessarily associated with episodic events.

Page 2, Lines 15-16: I would suggest adding the following cite following the Akritidis et al. (2010) reference: Lefohn et al. (2011). Lefohn, A.S., Wernli, H., Shadwick, D., Limbach, S.,

Oltmans, S.J., Shapiro, M., 2011. The importance of stratospheric-tropospheric transport in affecting surface ozone concentrations in the Western and Northern Tier of the United States. Atmospheric Environment 45, 4845-4857.

Page 2, Line 16: I would suggest changing "deep STT are usually rather weak with typical ozone enhancements of less…" to "deep STT can be rather weak with typical ozone enhancements of less…"

Page 2, Line 19: I would suggest changing "rarely" to "infrequently"

Page 2, Line 19: I would suggest changing "is most frequent" to "occurs typically."

Page 2, Line 19: I would suggest changing "manifold" to "as follows."

Page 2, Line 31: Please change "Jonson" to "Johnson."

Page 4, Line 4: I would suggest changing "accurate" to "correct".

Page 4, Line 27: I would suggest changing "setup" to "resolution and domain."

Page 5, Lines 3-6: The authors state the following: "The specific objectives of this numerical process study are: (1) to develop a complementary set of diagnostics that enables a quantitative investigation of stratospheric tracer transport by isentropic advection and turbulent mixing; (2) to better understand the role of boundary layer turbulence in transporting stratospheric air down to the surface; and (3) to obtain a rough estimate of maximum surface concentrations of stratospheric tracer in the two case studies." I would suggest in the paper identifying the absolute concentrations in units of ppb for Objective 3. Currently most of the discussion in the manuscript involves percentages instead of absolute concentrations. The reader can calculate the absolute concentrations from the assumed 200 ppb concentration times the percentage predicted near the surface, but I think it would be appropriate for the authors to clearly state their prediction in ppb.

Page 7, Line 8: I would suggest changing "in" to "for the."

Page 7, Line 9: I would suggest changing "chosen" to "selected."

Page 7, Line 17: I would suggest changing "had not been" to "was not."

Page 9, Lines 11-18: The authors state "Hence, locally, the frontal ageostrophic circulation most likely brings stratospheric tracer from the free troposphere (around 700 hPa) down to the surface after about 13 UTC 2 May 2006, when the tracer covers a large fraction of the target region on 300K (Fig. 4c). But the most important transport mechanism remains the growth of the PBL during daytime. Indeed, in the afternoon of 2 May 2006, the PBL top reaches levels of roughly 600 hPa as can be seen from the vertical cross-section in Fig. 3f – keeping in mind that local time is UTC−6 h. The stratospheric tracer is entrained at the PBL top and transported to the surface by turbulent motions. Therefore, high concentrations up to 20% of the stratospheric tracer occur at near-surface levels in the second half of 2 May (Fig. 4f)." For the authors edification, there is

very good agreement between their results and the STT-S tracer analyses performed by Lefohn et al. (2011) for the same site. The figure below illustrates the relationship between hourly average ozone concentration and the number of stratospheric "hits" to the surface based on the trajectory analysis described in Lefohn et al. (2011). Besides the "hits" described in the figure below, STT-S "hits" occurred throughout the spring and summer and were statistically related to daily maximum hourly average ozone concentrations ≥ 50 ppb.

[Figure]

Please note that Yellowstone NP is UTC-7 h rather than the UTC-6 h as indicated in the manuscript. The data are reported as local standard time.

Page 9, Line 17: "Stratospheric is misspelled.

Page 10: Please change "is available" to "are available."

Page 10, Line 2: The authors note that "Unfortunately, no representative surface station data from the Plateau is (sic) available to validate this inference." As indicated in my General comments above, data do exist for Mt. Waliguan for the June 2006 period. While the Mt. Waliguan site is outside the target area, it is nearby. It may be possible for the authors to obtain the hourly average ozone data from the project officer if they wish to expand their target area.

Page 14, Line 5: The authors state "In the early morning of 14 June 2006, 5 at 00 UTC (local time is UTC + 7h)…". The UTC time is correct in comparison to local time. However, as a caution, the data that are recorded in China sometimes refer to Beijing time (UTC + 8 h), even though the location may be different than the UTC + 8 h time zone. This is mentioned to the authors if they decide to request and use the Mt. Waliguan ozone data.

Page 18, Line 7: I would suggest changing "For an STT event over the Rocky Mountains and another one over" to "For an STT event over Yellowstone National Park and another one over".

Page 18, Line 24: The authors state "It is thus plausible that such events can lead to enhancements of surface ozone concentration by up to 50 ppbv." Is this a general statement or based on the results associated with the confirmed Yellowstone NP observations and the unconfirmed results associated with the Tibetan Plateau area? If this is a generalization, I would appreciate it if further documentation can be provided.

**References**

Ding, A., Wang, T., 2006. Influence of stratosphere-to-troposphere exchange on the seasonal cycle of surface ozone at Mount Waliguan in western China, Geophys. Res. Lett., 33, L03803, doi:10.1029/2005GL024760.

Ma, J., Tang, J., Zhou, X., Zhang, X., 2002. Estimates of the chemical budget for ozone at Waliguan observatory, J. Atmos. Chem., 41, 21–48, doi:10.1023/A:1013892308983, 2002.

Zhu, B., Akimoto, H., Wang, Z., Sudo, K., Tang, J., Uno, I, 2004. Why does surface ozone peak in summertime at Waliguan?, Geophys. Res. Lett., 31, L17104, doi:10.1029/2004GL020609.

---

## Author Comment (AC1) · 21 Mar 2019

Paper submitted to ACP

**"A numerical process study on the rapid transport of stratospheric air down to the surface over western North America and the Tibetan Plateau"**

Bojan Škerlak, Stephan Pfahl, Michael Sprenger, and Heini Wernli

**Response to the Reviewers' comments:**

We thank the reviewers, Sue Gray and Allen Lefohn, for their constructive comments that helped improving the manuscript. The following are our main changes in the revised version:

- Figures 3-6 have been redone, now including labels for selected geopotential height contours and showing extended cross sections for the Tibetan Plateau case, as suggested by S. Gray.
- The statement that "deep STT is comparatively rare" in our original manuscript turned out to be misleading and A. Lefohn commented in detail on the importance of deep STT for surface ozone enhancements. We now make clear (or even clearer than in the original version) that we have absolutely no objection to this perspective and we regard our study as fully compatible with studies emphasizing the important role of deep STT for surface ozone.
- We also clarified that our study should be regarded as a process study with an idealized tracer, and not as a "prediction" of surface ozone enhancements. For such a prediction one would need a model with a realistic initialization of ozone and preferentially with tropospheric chemistry (as already noted in the original version of our paper).

Below are the detailed replies (in blue) to the individual comments (in black).

**Reviewer 1 (Sue Gray)**

General comments:
This is a very nice paper that presents a relatively high-resolution modelling study (7 km horizontal grid spacing) of two cases of stratosphere to troposphere transport. In both cases stratospheric air descends down to the planetary boundary layer though the mechanisms leading to that descent differ for the two cases. The paper is clearly and concisely written and the results support the conclusions drawn. I recommend it for publishing subject to consideration of the following minor comments.

Specific comments:

1. Abstract p1 L16: Here it says that that the surface concentrations reach peak values of 10-20% in both cases which raises the question '% of what'. Add 'of the imposed stratospheric value' or similar.

We added "… 10-20% of the imposed stratospheric value …".

2. Although the focus of the study is on the processes leading to the transport of air from the stratosphere down to the boundary layer, rather than on the concentrations of tracer reached in the boundary layer, the paper would benefit from comparison of the concentrations found with those found in some of the comparable (i.e. using passive tracers) modelling studies referenced in the introduction to the paper. The papers referenced don't focus on transport down to the boundary layer, but it might be useful to compare the transport into the upper-troposphere with your study since this transport is an essential pre-requisite to the deeper transport.

We looked at peak concentrations reported in comparable passive tracer studies (Kowol-Santen and Ancellet, 2000; Gray, 2003, 2006) and include a brief discussion of this comparison in the revised version.

3. Robustness of transport across tropopause: Related to the above comment, it would be useful to include some discussion as to the potential issues with the realism of the transport across the tropopause surface as this is likely to be strongly dependent on the vertical resolution and on the implicit (and any explicit) diffusion in the model (which may or may not be realistic). Presumably the tracer is initialised as a step function across the tropopause - the model would be unlikely to be able to retain this step function even in the absence of any 'physically realistic' transport. How does the tracer advection scheme differ from the advection scheme used for prognostic variables in the model?

Thanks for this important remark. We agree that quantitively the downward transport of the stratospheric tracer depends on the definition of the tropopause, the tracer initialization (yes, we have chosen a step function across the 2-pvu tropopause), the advection scheme of the model, model parameterizations and model resolution. It would be far beyond the scope of this process-oriented study to do a systematic test of these sensitivities, but it is a good idea to include a brief discussion of the robustness of the results and the many factors that affect tracer transport in the model simulation in the revised version. For tracer advection, we used the same scheme that is implemented in COSMO for water species (Roches and Fuhrer 2012), a second-order mass-conserving and positive-definite scheme developed by Bott (1989). We mention this in the revised version.
Bott, A., 1989: A positive definite advection scheme obtained by nonlinear renormalization of the advective fluxes, Mon. Weather Rev. 117, 1006-1015.
Roches, A. and O. Fuhrer, 2012: Tracer module in the COSMO model. Tech. Rep. 20, Consortium for small-scale modelling.

4. Importance of convection: how important is convection in your case studies? The daytime growth of the boundary layer and associated turbulent mixing is key to transport in both cases, but there is no mention of the possible role of convective mixing (the type 4 mechanism in your Fig. 1). If convection does have a role in your case studies you might want to consider whether a convection-permitting simulation may yield more or less transport into the boundary layer. For example, we (Chagnon and Gray, 2010: https://doi.org/10.1029/2010JD014421) found relative insensitivity to the explicit representation of convection. There was a reduction in the spatial extent in the tracer transported into the lower-troposphere. In one of the cases, a case with mid-level convection, there was a substantial increase in the total tracer transported to lower tropospheric levels with explicit convection (but very little difference for the other two cases).

The convection scheme is not triggered in our simulations of the two cases and therefore we did not discuss the role of the type 4 mechanism. In the revised version, we explicitly mention the fact that no convection occurs in our episodes, as well as the possibility that for other cases convective transport can play an important role.

Technical corrections:

1. p1 L21: change to 'studies have revealed'.

Corrected, thank you.

2. p2 L10: change to 'question of whether'.

Corrected, thank you.

3. p5, L11: The resolution will be several times (∼6) that of the grid spacing which is the value you give here.

We agree and changed "resolution" to "grid spacing".

4. Fig 3-6: Labelling of geopotential height contours. I appreciate that the contour interval is included in the caption, but there is no indication of actual values. Can at least a few of the contours be labelled please?

Now a few contours are labelled such that the reader can identify the actual values of the geopotential height contours.

5. Fig 5, right column panels: It would be nice if the domains of the cross sections could be extended to the left so that the start region of the layer of high stratospheric tracer concentrations could be determined.

Thank you for this suggestion. We extended the domains of the cross sections towards the W/SW.

6. p11, L19: Please move the statement about the relationship of local and UTC time from p14 L4 to here (where the text refers to the 'afternoon of 14 June').

We now mention the relationship between local time and UTC already on p. 11.

7. p14, L32: correct typo 'advecction'.

Corrected, thank you.

8. p18, L10: add 'transport' after 'vertical'.

"transport" added.

9. p18. Consider hyphenating 'horizontally-averaged' and 'vertically-confined' to aid readability.

Hyphens added.

**Reviewer 2 (Allen Lefohn)**

General
Ozone contributions originating from the stratosphere can affect surface ozone concentrations by presenting themselves as episodic events (i.e., high hourly averaged concentrations for a relatively short period of time) or as enhanced surface concentrations (i.e., moderate hourly average enhancements sometimes over a short time or over a longer time period). The authors have focused on investigating episodic ozone events which occurred at two different locations (i.e., Yellowstone National Park (NP) and the Tibetan Plateau) using a passive stratospheric air mass tracer in a mesoscale model to explore the processes that enable the transport down to the surface. The events occurred in early May 2006 at Yellowstone NP and in mid-June 2006 on the Tibetan Plateau. In both cases, a tropopause fold associated with an upper-level front enabled stratospheric air to enter the troposphere. Despite the strongly differing dynamical processes at the two locations, stratospheric tracer concentrations at the surface reached peak values of 10-20% (i.e., 20-40 ppb) that added to the surface concentrations, corroborating the potential of deep stratosphere-to-troposphere transport events to episodically influence surface ozone concentrations in these two regions.

The authors noted 3 limitations of their study were (1) the low number of investigated STT events, (2) the negligence of tropospheric chemistry, and (3) the missing link to surface

observations. While the authors were able to compare their model predictions with the recorded hourly ozone data at Yellowstone NP, the authors indicated that no representative surface station data were available from the Tibetan Plateau. Therefore, while comparing their predictions with data from Yellowstone NP, they were unable to validate their predictions for the Tibetan Plateau. Recognizing this problem, they focused their attention on identifying and understanding the processes leading to the surface signals of the stratospheric tracer in the model simulation.

Here we would like to clarify an important misunderstanding: we don't regard our simulations as "predictions" of the influence of STT events on surface ozone. Making an accurate prediction has never been the intention of this study, and we would choose a completely different model setup if we aimed at making such a prediction (a mesoscale atmospheric chemistry model with a realistically initialized ozone field like, e.g., WRF-Chem). Also, of course, if our simulations were predictions then we would need to choose events for which sufficient observations are available to validate the predictions. However, from the beginning, the objective of our study has been to do an idealized tracer experiment (initialized with a step function from 0 to 1 across the tropopause) to study the dynamical processes involved in the tracer transport down to the surface. Given this objective, the fact that no observations are available to "validate" our results is not a major limitation (see also replies below).

I do have two concerns that I believe the authors should address. My first concern deals with the statement that the number of "deep STT" events is rare. The use of the term "deep STT" is used differently by researchers who have published in the literature.

Thank you for this important remark. We think that our use of the term "rare" might have been misleading (not so much the use of "deep STT"). With "rare" we did not want to imply that deep STT is not relevant for understanding surface ozone variations; what we meant is that the STT mass flux into the boundary layer ("deep STT" in Škerlak et al. 2014, see their Fig. 5) is substantially smaller than the STT mass flux across 500 or 700 hPa (Fig. 4 in Škerlak et al. 2014). For instance, over Central Europe, the STT mass flux across 500 hPa is about 100 kg km$^{-2}$ s$^{-1}$, across 700 hPa about 40 kg km$^{-2}$ s$^{-1}$, and into the boundary layer about 10 kg km$^{-2}$ s$^{-1}$ in the annual mean. With "rare" we wanted to emphasize this decrease of the STT mass flux by about an order of magnitude between the mid-troposphere and the boundary layer. But by no means, we attempted to imply that the STT mass flux into the boundary layer is insignificant. We improve this paragraph in the revised version to avoid such a misunderstanding.

In the manuscript, the authors used the term "deep STT" as discussed in Škerlak et al. (2014). Škerlak et al. (2014) defined "deep STE" as stratospheric air that reaches the PBL within 4 days or vice versa. The definition of "deep STT" (or STE) in the manuscript focuses on the processes required to transport the stratospheric air down to the PBL over a very short period and then down to the surface. Škerlak et al. (2014) note that for a subset of deep STE events, the downward ozone flux into the PBL is dominated by the mass flux and are most frequent in early spring. The authors concluded that surface ozone concentration along the west coast of North

America and around the Tibetan Plateau are likely to be influenced by deep stratospheric intrusions.

As noted by the authors, Lefohn et al. (2011 and 2012) described the influence of STT trajectories on surface ozone at Yellowstone NP. In their papers, Lefohn et al. (2011 and 2012) discuss the frequency of deep STT "hits" that are associated with periods greater than the 4-day criterion applied by Škerlak et al. (2014). Thus, I believe there is some confusion about the use of the term "deep STT". While the current manuscript refers to the rare occurrences of "deep STT", other papers indicate a greater number of occurrences of "deep STT" using a different set of criteria for the term "deep STT".

We don't think that the use of the term "deep STT" by Lefohn et al., Škerlak et al., and this study are inconsistent. Both Lefohn et al. and Škerlak et al. investigated STT trajectories, and then used different approaches to assess whether these trajectories "affected" the surface (Škerlak et al. used the reanalysis boundary layer height, and Lefohn et al. used a pragmatic threshold of static stability between the surface and the trajectory). We regard these approaches are equally meaningful and don't see a confusion. However, as mentioned above, we agree that our use of the term "rare" was confusing (corrected in the revised version).

I am not necessarily convinced that the "deep STT" events are as rare as the authors indicate in their manuscript if a broader definition of the term "deep STT" is applied. It is obvious that every "deep STT" event will not necessarily lead to an episodic enhancement of ozone (e.g., 20-40 ppb) at the surface. For example, Lefohn et al. (2011 and 2012) quantified the number of STT-S "hits" that occurred at the Yellowstone NP site. Lefohn et al. (2011, 2012 used a subjective criterion for selecting deep STT trajectories that potentially affect the surface by requesting that the potential temperature of the trajectory is less than 5K warmer than surface potential temperature. Their rationale was that a small vertical gradient of potential temperature can be overcome by boundary layer turbulence. While Lefohn et al. (2011 and 2012) were unable to quantify the turbulent transport down to the surface or study the involved processes, they did quantify the number of STT trajectories (STT-S) that were predicted to reach the surface and potentially enhance surface ozone concentrations. Lefohn et al. (2011) noted for sites across the US the following:

1. For the time period of their analyses, the high-elevation site at Yellowstone National Park (NP) in Wyoming exhibited more than 19 days a month during the spring and summer for hourly average ozone concentrations ≥ 50 ppb with STT-S> 0, where STT-S represented the number of deep trajectories. At this site, the daily maximum hourly springtime average ozone concentrations were usually in the 60-70 ppb range. The maximum daily 8-h average concentrations mostly ranged from 50 to 65 ppb;

2. At many of the lower-elevation sites, there was a preference for ozone enhancements to be coincident with STT-S> 0 during the springtime, although summertime occurrences were sometimes observed; and

3. For many cases, the coincidences between the enhancements and the STT-S events occurred over a continuous multiday period. When statistically significant coincidences occurred, the daily maximum hourly average concentrations were mostly in the 50-65 ppb range and the daily maximum 8-h average concentrations were usually in the 50-62 ppb range;

Lefohn et al. (2012) also noted:

1. STT down to the surface (STT-S) frequently contributed to enhanced surface ozone hourly averaged concentrations at sites across the US, with substantial year-to-year variability;

2. The ozone concentrations associated with the STT-S events appeared to be large enough to enhance the measured ozone concentrations during specific months of the year;

3. Months with a statistically significant coincidence between enhanced ozone concentrations and STT-S occur most frequently at the high-elevation sites in the Intermountain West, as well as at the high-elevation sites in the West and East;

4. These sites exhibit a preference for coincidences during the springtime and in some cases, the summer, fall, and late winter; and

5. Besides the high-elevation monitoring sites, low-elevation monitoring sites across the entire US experienced enhanced ozone concentrations coincident with STT-S events.
As indicate above, the number of STT-S "hits" as described by Lefohn et al. (2011, 2012) were frequent at many sites. For example, for the Yellowstone NP site in 2006, the figure below illustrates the daily maximum 8-h observed concentration and the number of daily STT-S "hits" over the entire year.

While the number of STT-S "hits" were frequently found at high-elevation sites in the US during the spring and summer, they were also found at times during the springtime at low-elevation sites. The number of STT-S hits were mostly associated with subtle enhancements of surface ozone concentrations when compared to the number of episodic events. In other words, many of the "deep STT" events were associated with subtle enhancements to surface ozone, while other less frequent "deep STT" events were associated with episodic additions to the surface levels. In summary, "deep STT" events contributed to the more frequent enhancements of surface ozone concentrations rather than the episodic events that raised surface ozone levels (20-40 ppb).

We completely agree with this detailed summary of the important findings of the Lefohn et al. studies. And again, we apologize for our misleading statement that "deep STT are comparatively

rare" – this was not meant to question the results by Lefohn et al. who demonstrated in detail that deep STT strongly affects surface ozone in many parts of North America.

I would recommend that the authors carefully define in their manuscript the term "deep STT" and caution the reader that other researchers have reported deep STT events using different criteria than those used by the authors. Clearly, other published works have discussed the importance of stratospheric ozone transport into the PBL to shape the distribution of hourly average concentrations at both high- and low-elevation sites. For example, the Mt. Waliguan site on the Tibetan Plateau is highly representative of free-tropospheric ozone (Ma et al., 2002) and is often influenced by stratosphere-to-troposphere transport (STT) events (Ding and Wang, 2006; Zhu et al., 2004).

Perhaps the authors should expand their discussion in the Introduction to include their views on the importance of STT processes that influence surface ozone concentrations that include both episodic, as well as subtle enhancements. I believe a more balanced discussion should be considered.

We revise our introduction to make clear that we are fully in line with the many studies that show an important impact of deep STT on surface ozone, and we more carefully explain what we meant by "comparatively rare".

My second concern is associated with the inability of the authors to validate their predictions with a site in the Tibetan Plateau. Hourly ozone data for Mt. Waliguan, China (Latitude 36° 17' N; Longitude 100° 54' E) have been recorded for June 2006. The authors might wish to obtain the hourly ozone data by requesting the information for this time period by contacting Dr. Xiaobin Xu (Key Laboratory for Atmospheric Chemistry, Institute of Atmospheric Composition, Chinese Academy of Meteorological Sciences, China Meteorological Administration, Zhongguancun Nandajie 46, Beijing 100081, China - xiaobin_xu@189.cn). On 17 June 2006 at 0300 Beijing Time (UTC + 8 hours), the hourly average ozone concentration was similar in magnitude to the episode that occurred at Yellowstone NP described in the manuscript. The site is situated at the northeastern edge of the Tibetan Plateau. While the Mt. Waliguan site is outside of the target region defined in the manuscript, the authors might wish to modify their analyses so that their predictions can be compared with actual ozone data recorded in the Tibetan Plateau for the June 2006 period. This is a decision I will leave to the authors and perhaps the editor.

We reemphasize that our simulation results should not be regarded as predictions. We carefully checked for available surface ozone measurements that could be meaningfully compared (in a qualitative sense) with our tracer results, but we failed. Mt. Waliguan is clearly too far away to be affected by our event (Fig. 6f shows that our tracer reaches high surface concentrations between 80°E and 90°E, but Mt. Waliguan is at 100°E). When writing the paper, we also contacted Prof. Shichang Kang, who operates the Nam Co Station in Tibet (30° 42′ N, 90° 33′ E), but unfortunately, they started ozone measurements only in October 2010. As we have mentioned in

the paper, this lack of surface observations is a caveat of our study, however, in our view, this does not reduce the value of our model-based process-oriented study on the understanding of how stratospheric air can be transported down to the surface. We regard our contribution as complementary to the many and important observation-based studies on the influence of deep STT on surface ozone.

I would recommend that the manuscript be published once my first concern is addressed. As indicated above, my second concern about the addition of the Mt. Waliguan data to their analyses is a decision left to the authors and perhaps the editor.

Specific Suggestions

We are very grateful for the detailed and specific (language) suggestions!

Abstract

Page 1, Line 2-3: I suggest changing "these plumes of" to "mid-June plumes associated with".

We keep the original formulation since this statement is meant to be general and not restricted to STT events that occur in June.

Page 1, Line 4-6: I suggest changing "(ii) if boundary layer turbulence is strong enough to enable this transport, the originally stratospheric air mass is strongly diluted by mixing such that only a weak stratospheric signal can be recorded at the surface" to "(ii) even if boundary layer turbulence were strong enough to enable this transport, the originally stratospheric air mass can be diluted by mixing, such that only a weak stratospheric signal can be recorded at the surface."

Thank you, we changed to the formulation as suggested.

Page 1, Line 6: I suggest changing "from" to "associated with."

Changed as suggested.

Page 1, Lines 8-9: I suggest changing "The events occurred in early May 2006 in the Rocky Mountains and in mid June 2006 on the Tibetan Plateau" to "The events occurred in early May 2006 at Yellowstone National Park in Wyoming and in mid-June 2006 on the Tibetan Plateau."

Figure 4f shows that a large area, extending over more than 1000 km in the W-E direction, is affected by the stratospheric tracer at the surface. It would therefore be misleading to label this event as one that occurred specifically at Yellowstone National Park.

Page 1, Line 12: I suggest changing "and a reservoir" to "and initially a reservoir."

Changed as suggested.

Page 1, Line 13: I suggest changing "entrainment" to "However, entrainment..."

Changed as suggested.

Page 1, Line 14: I suggest changing "fosters" to "allows for..."

Changed as suggested.

Page 1, Line 15: I suggest removing the word "Interestingly" and starting the sentence with "Despite."

Changed as suggested.

Page 1, Line 16: Would it be possible to place into parenthesis the range of ozone concentrations at the surface such as 10-20% (i.e., 20-40 ppb)" or whatever the range of concentrations is?

We prefer not to do so because the 10-20% are solid results from our simulations, whereas the "translation" to "20-40 ppbv" requires a stratospheric ozone concentration, which is somehow subjective (see also reply below on next page to comment "Page 5, Lines 3-6").

Introduction

Page 1, Line 19: I suggest replacing "known" with "documented."

Changed as suggested.

Page 2, Line 1: I would suggest citing Langford et al. (2009). Langford, A.O., Aikin, K.C., Eubank, C.S., Williams, E.J., 2009. Stratospheric contribution to high surface ozone in Colorado during springtime. Geophysical Research Letters 36, L12801. http://dx.doi.org/10.1029/2009GL038367.

Thanks for the suggestion, the reference has been added.

Page 2, Lines 4-5: I would suggest changing "cross the tropopause and then to descend down to" to "cross the tropopause and to descend to".

Changed as suggested.

Page 2, Lines 5-6: I would suggest changing "This isentropic descent typically goes along with" to "This isentropic descent typically is accompanied with".

Changed as suggested.

Page 2, Line 12: I would suggest changing "that deep STT events are comparatively rare" to "that deep STT events are comparatively infrequent..." (Again, I would suggest that the authors re-evaluate how "rare "deep STT" events are. Several authors have reported frequent "deep STT" events that are not necessarily associated with episodic events.

We carefully reformulated this sentence. It now reads "STT events are less frequent than STT events into the mid-troposphere".

Page 2, Lines 15-16: I would suggest adding the following cite following the Akritidis et al. (2010) reference: Lefohn et al. (2011). Lefohn, A.S., Wernli, H., Shadwick, D., Limbach, S., Oltmans, S.J., Shapiro, M., 2011. The importance of stratospheric-tropospheric transport in affecting surface ozone concentrations in the Western and Northern Tier of the United States. Atmospheric Environment 45, 4845-4857.

The reference has been added.

Page 2, Line 16: I would suggest changing "deep STT are usually rather weak with typical ozone enhancements of less..." to "deep STT can be rather weak with typical ozone enhancements of less..."

Changed as suggested.

Page 2, Line 19: I would suggest changing "rarely" to "infrequently"

Changed as suggested.

Page 2, Line 19: I would suggest changing "is most frequent" to "occurs typically."

Changed as suggested.

Page 2, Line 19: I would suggest changing "manifold" to "as follows."

Changed as suggested.

Page 2, Line 31: Please change "Jonson" to "Johnson."

Thank you, typo has been corrected.

Page 4, Line 4: I would suggest changing "accurate" to "correct".

Changed as suggested.

Page 4, Line 27: I would suggest changing "setup" to "resolution and domain."

We kept "setup" because it includes also the choice of the idealized tracer as mentioned in the previous sentence.

Page 5, Lines 3-6: The authors state the following: "The specific objectives of this numerical process study are: (1) to develop a complementary set of diagnostics that enables a quantitative investigation of stratospheric tracer transport by isentropic advection and turbulent mixing; (2) to better understand the role of boundary layer turbulence in transporting stratospheric air down to the surface; and (3) to obtain a rough estimate of maximum surface concentrations of stratospheric tracer in the two case studies." I would suggest in the paper identifying the absolute concentrations in units of ppb for Objective 3. Currently most of the discussion in the manuscript involves percentages instead of absolute concentrations. The reader can calculate the absolute concentrations from the assumed 200 ppb concentration times the percentage predicted near the surface, but I think it would be appropriate for the authors to clearly state their prediction in ppb.

We prefer not to follow this suggestion. Assuming a stratospheric concentration of 200 ppb is somehow arbitrary and providing an absolute concentration value at the surface might give the misleading impression that we aim at predicting surface ozone enhancements for the two cases (which is not the intention of this paper, as stated above). In the conclusion, we write "For both STT events … peak surface concentrations of the stratospheric tracer reached values of about 20%. It is thus plausible that such events can lead to enhancements of surface ozone concentration by up to 50 ppbv." We think that this is an appropriate statement about absolute surface ozone concentrations in such an idealized tracer study.

Page 7, Line 8: I would suggest changing "in" to "for the."

We kept "in" ("in the model").

Page 7, Line 9: I would suggest changing "chosen" to "selected."

We simplified the sentence.

Page 7, Line 17: I would suggest changing "had not been" to "was not."

Changed as suggested.

Page 9, Lines 11-18: The authors state "Hence, locally, the frontal ageostrophic circulation most likely brings stratospheric tracer from the free troposphere (around 700 hPa) down to the surface after about 13 UTC 2 May 2006, when the tracer covers a large fraction of the target region on 300 K (Fig. 4c). But the most important transport mechanism remains the growth of the PBL during daytime. Indeed, in the afternoon of 2 May 2006, the PBL top reaches levels of roughly 600 hPa as can be seen from the vertical cross-section in Fig. 3f – keeping in mind that local time is UTC−6 h. The stratospheric tracer is entrained at the PBL top and transported to the surface by turbulent motions. Therefore, high concentrations up to 20% of the stratospheric tracer occur at near-surface levels in the second half of 2 May (Fig. 4f)." For the authors edification, there is very good agreement between their results and the STT-S tracer analyses performed by Lefohn et al. (2011) for the same site. The figure below illustrates the relationship between hourly average ozone concentration and the number of stratospheric "hits" to the surface based on the trajectory analysis described in Lefohn et al. (2011). Besides the "hits" described in the figure below, STT-S "hits" occurred throughout the spring and summer and were statistically related to daily maximum hourly average ozone concentrations ≥ 50 ppb.

We added the following sentence: "These model-based results are fully in line with the trajectory-based analysis of the same STT episode in Lefohn et al. (2011)."

Please note that Yellowstone NP is UTC-7 h rather than the UTC-6 h as indicated in the manuscript. The data are reported as local standard time.

Thank you, we corrected this mistake.

Page 9, Line 17: "Stratospheric" is misspelled.

Thank you, typo has been corrected.

Page 10: Please change "is available" to "are available."

Page 10, Line 2: The authors note that "Unfortunately, no representative surface station data from the Plateau is (sic) available to validate this inference." As indicated in my General comments above, data do exist for Mt. Waliguan for the June 2006 period. While the Mt. Waliguan site is outside the target area, it is nearby. It may be possible for the authors to obtain the hourly average ozone data from the project officer if they wish to expand their target area.

Unfortunately, the Mt. Waliguan station is too far away from the simulated surface ozone peak and therefore does not provide additional information for our event.

Page 14, Line 5: The authors state "In the early morning of 14 June 2006, 5 at 00 UTC (local time is UTC + 7h) ...". The UTC time is correct in comparison to local time. However, as a caution, the data that are recorded in China sometimes refer to Beijing time (UTC + 8 h), even

though the location may be different than the UTC + 8 h time zone. This is mentioned to the authors if they decide to request and use the Mt. Waliguan ozone data.

See above.

Page 18, Line 7: I would suggest changing "For an STT event over the Rocky Mountains and another one over" to "For an STT event over Yellowstone National Park and another one over".

Figure 4f shows that a large area, extending over more than 1000 km in the W-E direction, is affected by the stratospheric tracer at the surface. It would therefore be misleading to label this event as one that occurred specifically over the Yellowstone National Park.

Page 18, Line 24: The authors state "It is thus plausible that such events can lead to enhancements of surface ozone concentration by up to 50 ppbv." Is this a general statement or based on the results associated with the confirmed Yellowstone NP observations and the unconfirmed results associated with the Tibetan Plateau area? If this is a generalization, I would appreciate it if further documentation can be provided.

This is meant as a very general and carefully formulated statement ("it is plausible"). If 20% of the stratospheric tracer can reach the surface, then assuming a stratospheric ozone concentration of 250 ppbv (you suggested 200 ppbv above) would lead to a surface ozone enhancement up to 50 ppbv. Such enhancements have been reported in the literature (from observational studies) and it is nice to see that our idealized experiments – although only for two specific STT episodes – support such strong enhancements.